# Wintertime secondary organic aerosol formation in Beijing-Tianjin-Hebei (BTH): Contributions of HONO sources and heterogeneous reactions

Li Xing[1], Jiarui Wu[1], Miriam Elser[2], Shengrui Tong[3], Suixin Liu[1], Xia Li[1], Lang Liu[1], Junji Cao[1*], Jiamao Zhou[1], Imad El-Haddad[2], Rujin Huang[1], Maofa Ge[3], Xuexi Tie[1], André S. H. Prévôt[2], and Guohui Li[1*]

[1]Key Lab of Aerosol Chemistry and Physics, SKLLQG, Institute of Earth Environment, Chinese Academy of Sciences, Xi'an, China
[2]Laboratory of Atmospheric Chemistry, Paul Scherrer Institute, 5232 Villigen, Switzerland
[3]State Key Laboratory for Structural Chemistry of Unstable and Stable Species, Beijing National Laboratory for Molecular Sciences (BNLMS), Institute of Chemistry, Chinese Academy of Sciences, Beijing, China

*Correspondence to: Guohui Li (ligh@ieecas.cn) and Junji Cao (jjcao@ieecas.cn )

**Abstract**: Organic aerosol (OA) concentrations are simulated over the Beijing-Tianjin-Hebei (BTH) region from 9 to 26 January, 2014 using the Weather Research and Forecasting model coupled with chemistry (WRF-CHEM) model, with the goal of examining the impact of heterogeneous HONO sources on SOA formation and the SOA formation from different pathways during wintertime haze days. The model generally shows a good performance in simulating air pollutants and organic aerosols against measurements in BTH. Model results show that heterogeneous HONO sources substantially enhance the near-surface SOA formation, increasing regional average near-surface SOA concentration by about 46.3% during the episode. Oxidation and partitioning of primary organic aerosols treated as semi-volatile dominate the SOA formation, contributing 58.9% of the near-surface SOA mass in BTH. Irreversible uptake of glyoxal and methylglyoxal on aerosol surfaces constitutes the second most important SOA formation pathway during the episode, with SOA contribution increasing from 8.5% in non-haze conditions to 30.2% in haze conditions. Additionally, direct emissions of glyoxal and methylglyoxal from residential living sources contribute about 25.5% to the total SOA mass on average in BTH. Our study highlights the importance of heterogeneous HONO sources and primary residential emissions of glyoxal and methylglyoxal to SOA formation in winter over BTH.

# 1      Introduction

Organic aerosols (OA) are one of the most important components of fine particulate matters ($PM_{2.5}$), constituting 20%-90% of $PM_{2.5}$ mass in the northern hemisphere (Zhang et al., 2007). OA not only scatter or absorb a fraction of the incoming solar radiation, also serve as cloud condensation nuclei and ice nuclei, directly and indirectly influencing the radiative energy budget of the Earth-atmosphere system (IPCC, 2013). OA are generally classified into two types: primary OA (POA) and secondary OA (SOA). POA are directly emitted into the atmosphere as particles by various anthropogenic

and biomass burning sources, while SOA are formed from the complex oxidation of volatile organic compounds (VOCs) followed by gas-particle transfer processes or heterogeneous reactions of carbonyls. Some species of POA evaporate into the atmosphere and are oxidized further, re-partition into aerosols, and form SOA (Robinson et al., 2007; Hallquist et al., 2009; Shrivastava et al., 2017).

      China has been suffering from severe haze pollution in winter within these recent years, especially over the

Beijing-Tianjin-Hebei (BTH) region (Guo et al., 2014; Bei et al., 2016; Chang et al., 2016). Observations have shown that OA play a critical role in the haze pollution in China (Xing et al., 2013; Sun et al., 2013; Huang et al., 2014; Li et al., 2017). Huang et al. (2014) have reported that OA account for 30%-50% of the $PM_{2.5}$ mass in four megacities of China during severe haze days, with the SOA contribution ranging from 44% to 71% in winter. Sun et al. (2013) have observed that OA contribute 52% of the non-refractory submicron aerosol ($NR-PM_1$) in Beijing in the winter of 2012 and SOA constitute 31%

of the OA mass. H. Li et al. (2017) have found that OA dominate the $PM_1$ mass during wintertime heavy haze days in Handan, China and SOA make up 39% of the total OA mass.

      The hydroxyl radical (OH) is one of the most important oxidants in the troposphere, controlling the daytime atmospheric oxidation capacity (AOC) and further affecting ozone ($O_3$) and SOA formations (Volkamer et al., 2010; Stone et al., 2012). G. Li et al. (2017) have demonstrated that the $O_3$ concentration is fairly low when $PM_{2.5}$ concentrations are higher

than 200 μg m$^{-3}$ during wintertime in the Guanzhong Basin, China, revealing the limited AOC under severe haze pollution conditions. Meanwhile, the high contribution of SOA to the OA mass during severe haze days indicates that there exist other OH sources promoting the SOA formation via the oxidation of VOCs or enhanced heterogeneous reactions of carbonyls.

      Photolytically liable nitrous acid (HONO) is an important OH source, particularly during the early morning hours when the other OH sources are less important in the polluted atmosphere (Stutz et al., 2000; Li et al., 2010). Recent studies have

shown that the reaction of NO and OH cannot interpret the observed high HONO concentrations in both urban and rural areas (Li et al., 2010; Li et al., 2015). Heterogeneous sources have been considered to be significant for the atmospheric HONO formation, including direct emissions from vehicles, nitrogen dioxide ($NO_2$) heterogeneous reactions on aerosol and ground surfaces, and $NO_2$ reduction reactions with organics and soot (Arens et al., 2001; Gutzwiller et al., 2002; Aumont et al., 2003; Ndour et al., 2008). Several model studies have shown that including the HONO heterogeneous source has

reasonably reproduced the observed high HONO level and consequently enhance the simulated $O_3$ and SOA concentrations (Li et al., 2010; Li et al., 2015; Zhang et al., 2016). For example, Li et al. (2010) have shown that additional heterogeneous HONO sources elevate the simulated SOA concentration by a factor of 2 in the morning in Mexico City. Li et al. (2015) have revealed that additional HONO sources increase simulated $O_3$ and $PM_{2.5}$ concentrations by around 9 ppb and 32 μg m$^{-3}$ during daytime in August 2007 in BTH.

Heterogeneous reactions are also an important SOA formation pathway (Fu et al., 2008; Li et al., 2013). Laboratory and field studies have indicated that glyoxal and methylglyoxal cause the rapid SOA production via aerosol uptake or cloud processing (Liggio et al., 2005; Volkamer et al., 2007). Li et al. (2013) have included the aqueous uptake of glyoxal and methylglyoxal into wet aerosols and cloud droplets as an additional SOA formation pathway in the 3-dimensional regional air quality model CMAQ and simulated the SOA formation in the Pearl River Delta region of China. Simulations show that

the aqueous uptake of glyoxal and methylglyoxal helps to narrow the gap in SOA concentrations between models and measurements.

SOA simulations in chemical transport models (CTMs) have been substantially improved in recent years. Odum et al. (1996) have proposed a traditional two-product model to describe SOA production, in which two oxidation products with different saturation vapor pressures are produced from one specific VOC precursor oxidation and then reversibly partition

between the gas and particle phase to form SOA. The two-product model has been widely used in CTMs to simulate SOA formation, although it generally tends to underestimate SOA concentrations (Chung and Seinfeld, 2002; Henze and Seinfeld, 2006). Donahue et al. (2006) have proposed a volatility basis-set (VBS) approach to represent the wide range volatilities of organic species and the aging of SOA can be easily represented by the mass transfer among different volatility bins. CTMs using the VBS approach have remarkably improved the SOA simulations against observations (e.g., Li et al., 2011;

Shrivastava et al., 2013, 2015; Feng et al., 2016).

Previous studies have investigated the OA formation in China and generally tend to underestimate OA concentrations (Han et al., 2008, 2016; Fu et al., 2012; Fu and Liao, 2012; Jiang et al., 2012; Li et al., 2013; Tsai et al., 2015; Feng et al., 2016; Chen et al., 2017; Hu et al., 2017). Jiang et al. (2012) have used the two-product model to simulate SOA in 2006 in China and found that the model underestimates SOA concentrations by 0-75%. Fu et al. (2012) have simulated organic

carbon (OC) in China using the two-product model with the aqueous uptake of glyoxal and methylglyoxal, showing that the model significantly underestimates the observed OC concentrations in all seasons and fails to capture the spatiotemporal variability of OC. Han et al. (2016) have used the VBS approach and two-product model to simulate OA over East China in April 2009. The simulated SOA concentrations using VBS approach are higher than those using the two-product model. Additionally, the predicted ratio of secondary OC to total OC in the VBS approach is about 33%, much higher than that

(around 5%) in the two-product model and also close to observation-based estimation (32%), suggesting a more realistic representation of the SOA formation by the VBS approach through accounting for the semi-volatile and intermediate

volatility organics emitted from fossil fuel and biomass burning sources. Hu et al. (2017) have modeled SOA formation in China in 2013 using the two-product model and the simulation underestimates observed OC concentrations in the winter in Beijing, especially during heavy haze days. Recent studies have demonstrated that CTMs are subject to underestimating SOA concentrations against measurements using the traditional two-product SOA module, particularly during wintertime haze days with rather low $O_3$ level (e.g., Jiang et al., 2012; Fu et al., 2012; Hu et al., 2017). Hence, it is imperative to improve the SOA simulations for supporting the design and implementation of emission control strategies to mitigate haze pollution in China.

In the study, the VBS SOA approach with aging implemented in the WRF-CHEM model is used to attempt to improve the SOA simulation during wintertime haze days in BTH, with focuses on the contribution of the heterogeneous HONO sources and the uptake of glyoxal and methylglyoxal to the SOA formation. The WRF-CHEM model configuration and observation data are described in section 2. The model results are analyzed in section 3. The conclusions are summarized in section 4.

## 2 Model configuration and observation data

### 2.1 WRF-CHEM model

The version of the WRF-CHEM model is developed by Li et al. (2010) and the OA module used in this study is incorporated into the model by Li et al. (2011). Briefly, the model uses the SAPRC-99 gas-phase chemical mechanism, with the aerosol module in CMAQ/Models3 developed by US EPA (Binkowski and Roselle, 2003). Three different modes of log-normal distributions are superposed to represent the aerosol size distribution. Particle nucleation, coagulation, and size growth/shrink by the addition/loss of mass are included in the aerosol module. The photolysis rates of gas phase species are calculated by the Fast Tropospheric Ultraviolet and Visible (FTUV) Radiation Model considering the aerosol effects on photolysis frequencies (Tie et al., 2003; Li et al., 2005). Inorganic aerosols are simulated using the ISORROPIA version 1.7 (Nenes et al., 1998). The dry and wet deposition of chemical species are calculated using the parameterization by Wesely (1989) and the method in CMAQ/Models3, respectively.

The OA module is based on the VBS approach with aging and detailed information can be found in Li et al. (2011). The POA components from traffic-related combustion and biomass burning are represented by nine surrogate species with saturation concentrations ($C^*$) ranging from $10^{-2}$ to $10^6$ µg m$^{-3}$ at room temperature (Shrivastava et al., 2008), and assumed to be semi-volatile and photochemically reactive (Robinson et al., 2007). The SOA formation from each anthropogenic or biogenic precursor is calculated using four semi-volatile VOCs with effective saturation concentrations of 1, 10, 100, and 1000 µg m$^{-3}$ at 298 K. Previous studies have demonstrated that the fragmentation reactions of semi-volatile VOCs also play an important role in the SOA formation (Shrivastava et al., 2013, 2015, 2016). However, the fragmentation reactions have

 The SOA formation via the heterogeneous reaction of glyoxal and methylglyoxal is parameterized as a first-order irreversible uptake by aerosol particles with an uptake coefficient of $3.7\times10^{-3}$ (Liggio et al., 2005; Zhao et al., 2006; Volkamer et al., 2007).

Besides the homogeneous formation HONO by the reaction of NO and OH, the heterogeneous HONO sources are also considered in the model, including secondary HONO formation from heterogeneous $NO_2$ reaction with semi-volatile organics and freshly emitted soot, and the heterogeneous reaction of $NO_2$ on aerosol and ground surfaces. Details about the model parameterization of the heterogeneous HONO formation can be found in Li et al. (2010).

## 2.2    Model configuration

The WRF-CHEM model is used to simulate a persistent air pollution episode occurred in BTH from 9 to 26 in January 2014 to investigate the SOA formation. The model is set up with a horizontal grid resolution of 6 kilometers and $150\times150$ grid cells centered at 39ºN and 117ºE (Figure 1). Thirty-five vertical levels are utilized with finer vertical resolution near the surface. The model employs the microphysical scheme of Hong and Lim (2006), the MYJ TKE planetary boundary layer scheme (Janjić, 2002), the MYJ surface layer scheme (Janjić, 2002), the Unified Noah land-surface model (Chen and Dudhia, 2001), and the Goddard shortwave and longwave radiation schemes (Chou and Suarez, 1999; 2001) in simulations. The NCEP $1º\times1º$ reanalysis data (https://rda.ucar.edu/datasets/ds083.2/) is used for the meteorological initial and boundary conditions. The Model for OZone And Related chemical Tracers (MOZART) output with 6 h interval (Horowitz et al., 2003) provides the chemical initial and boundary conditions. The spin-up time for initialization is 2 days.

The anthropogenic emission inventory used in the study includes agriculture, industry, power plant, residential, and transportation sectors in the base year of 2013 (Zhang et al., 2008; M. Li et al, 2017). The biogenic emissions are calculated online using the Model of Emissions of Gases and Aerosols from Nature (MEGAN) module (Guenther et al., 2006).

## 2.3    Observation data

The measurement data of hourly $PM_{2.5}$, $SO_2$, $NO_2$, and $O_3$ concentrations in BTH are downloaded from the website http://www.aqistudy.cn/ released by China's Ministry of Ecological Environment. OA have been measured by the Aerodyne High Resolution Time-of-Flight Aerosol Mass Spectrometer (HR-ToF-AMS) with a novel $PM_{2.5}$ lens (Williams et al., 2013) from 9 to 26 January 2014 at the Institute of Remote Sensing and Digital Earth (IRSDE), Chinese Academy of Sciences (40.00°N, 116.38°E) in Beijing (Figure 1). The Positive Matrix Factorization (PMF) technique is used to identify the OA sources (Canonaco et al., 2013; Elser et al., 2016). Five components of OA are classified by their mass spectra and time series, including traffic-combustion hydrocarbon-like OA (HOA), cooking OA (COA), biomass burning OA (BBOA), coal combustion OA (CCOA), and oxygenated OA (OOA). HOA, COA, BBOA, and CCOA are interpreted for surrogates of

primary OA (POA), and OOA is a surrogate for SOA. The details of the HR-ToF-AMS measurement and the source apportionment of OA can be found in Elser et al. (2016). HONO has also been measured by a homemade HONO analyzer at the IRSDE site. Further details about the measurement procedure can be found in Tong et al. (2016).

## 2.4 Statistical indexes for comparisons

The mean bias (MB), root mean square error (RMSE) and index of agreement (IOA) are used to assess the model prediction of aerosol species.

$$MB = \frac{1}{N}\sum_{i=1}^{N}(P_i - O_i)$$

$$RMSE = \left[\frac{1}{N}\sum_{i=1}^{N}(P_i - O_i)^2\right]^{\frac{1}{2}}$$

$$IOA = 1 - \frac{\sum_{i=1}^{N}(P_i - O_i)^2}{\sum_{i=1}^{N}(|P_i - \bar{O}| + |O_i - \bar{O}|)^2}$$

Where $P_i$ and $O_i$ are the simulated and observed concentrations of chemical species, respectively. $N$ is the number of model and observation data for comparisons. $\bar{O}$ is the average observed species concentration. *IOA* ranges from 0 to 1 and larger *IOA* indicates better agreement between model and observation.

## 3 Results and Discussions

In our previous study, the WRF-CHEM simulation of the haze pollution episodes has been validated using the air pollutant observations in BTH (Li et al., 2018). Generally, the model well predicts the horizontal distributions of $PM_{2.5}$, $O_3$, $NO_2$, and $SO_2$ mass concentrations against measurements. In addition, the model also reasonably well reproduces the temporal profiles of the air pollutants, but is subject to underestimation during the haze dissipation stage compared to observations.

The OA simulation is further compared with the HR-ToF-AMS data analyzed using PMF technique at IRSDE site in Beijing (Elser et al., 2016). The PMF results are called as "observation" hereafter, even if they are the model results constrained by observations. We have defined the base simulation including various anthropogenic and biogenic emission sources and the heterogeneous HONO formation as Li et al. (2010) (Henceforth referred to as BASE case), and results from the BASE case are compared with the observed POA and SOA in Beijing.

## 3.1 POA simulations

Figure 2 presents the temporal profiles of the simulated and observed POA (sum of HOA, BBOA, CCOA, and COA), HOA, BBOA+COA, and CCOA concentrations from 9 to 26 January 2014 at IRSDE site in Beijing. The model generally

yields the diurnal variations of the POA concentration compared to the HR-ToF-AMS measurements, with an *IOA* of 0.83 (Figure 2a). However, the model tends to overestimate the POA concentration, with a *MB* of 8.7 μg m⁻³, although it

frequently cannot reproduce the observed high peaks during heavy haze days. The POA simulation also exhibits rather large dispersions, with a *RMSE* of 35.5 μg m⁻³. It is worth noting that, the POA concentration in Beijing is dominated by primary emissions of vehicles, cooking, biomass burning, coal combustion, and trans-boundary transport from outside of Beijing, so uncertainties in emissions from various anthropogenic sources and the simulated meteorological fields substantially affect the simulated POA concentrations (Bei et al., 2017).

The model generally replicates the diurnal variations of HOA, BBOA+COA, and CCOA against the observations, with *IOAs* of 0.72, 0.69, and 0.81, respectively. The model fails to capture the peaks of all the POA components during the nighttime of 11 and 17 January 2014, which is likely caused by the emission uncertainty. The HOA simulation is slightly better than that of BBOA+COA. One of possible reasons is that the HOA emissions from vehicles have a more clear diurnal variation than those for BBOA and COA. Detailed discussions for the CCOA simulation can be found in Li et al. (2018).

**3.2    SOA simulations and HONO contributions**

Hydroxyl radical (OH), generally as an $O_3$ photochemical derivative, dominates the oxidation of VOCs and primary organic gases at daytime, affecting the SOA formation in the atmosphere. However, insolation in North China becomes weak during wintertime, which does not facilitate the $O_3$ formation. Low surface $O_3$ concentrations have been observed, particularly during heavy haze episodes, reducing the OH production from $O_3$ photolysis (G. Li et al., 2017). Photolytically

liable HONO has been reported to be a major OH source when the $O_3$ level is low, such as in the morning in urban areas (Li et al., 2010; Czader et al., 2012). Figure 3 shows the diurnal cycle of observed $O_3$ and HONO concentrations from 9 to 26 January 2014 at IRSDE site in Beijing. Apparently, the observed peak $O_3$ concentration is low, around 18 ppb, unfavorable for the photochemical production of OH. Therefore, an alternative compensation to the atmospheric OH is the observed high HONO level, with the lowest concentration of 0.75 ppb in the afternoon and up to 3.0 ppb during nighttime.

We further quantitatively evaluate the contribution of $O_3$ and HONO to the OH production based on the measurements at IRSDE site using the Tropospheric Ultraviolet and Visible (TUV) Radiation Model (http://cprm.acom.ucar.edu/Models/TUV/Interactive_TUV/). The calculation location is the IRSDE observation site (40.00°N, 116.38°E, Figure 1) and the time and date are 15:00 BJT and 15 January 2014, respectively. For the calculation of photolysis rates using the TUV model, the column $O_3$ is set to be 300 Dobson Unit and the aerosol and cloud effects are not

considered. $O_3$ photolysis generates the excited oxygen atom $O(^1D)$ and $O(^1D)$ reacts with water vapor to form OH:

$$O_3 + h\nu(\lambda < 310nm) \rightarrow O(^1D) + O_2 \qquad j_{O_3} = 3.2 \times 10^{-6}\ s^{-1}$$

$$O(^1D) + H_2O \rightarrow 2OH$$

However, the large majority (>90%) of $O(^1D)$ atoms are quenched to ground-state atoms $O(^3P)$ by collisions with nitrogen

and oxygen. Therefore, an upper limit estimation of OH production rate can be expressed as: $j_{O_3} \times [O_3] \times 0.1 \times 2$. Where $j_{O_3}$ is the $O_3$ photolysis rate and $[O_3]$ represents the $O_3$ mixing ratio. In Figure 3a, at 15:00 BJT, the $[O_3]$ is about 18 ppb, so the estimated maximal OH production rate from $O_3$ photolysis is about $1.2 \times 10^{-5}$ ppb s$^{-1}$.

HONO photolysis directly produces OH, but OH reacts with NO to reform HONO:

$$HONO + hv(300nm < \lambda < 405nm) \rightarrow OH + NO \qquad j_{HONO} = 6.8 \times 10^{-4} \, s^{-1}$$

$$OH + NO + M \rightarrow HONO + M$$

Model studies and measurements in Mexico City have shown that the contribution of the reaction of OH with NO to the HONO formation does not exceed 60% during daytime (Dusanter et al., 2009; Li et al., 2010). We use a lower limit that 20% of OH yielded from HONO photolysis does not recycle and the net OH production rate from HONO photolysis can be expressed as: $j_{HONO} \times [HONO] \times 0.2$. At 15:00 BJT, [HONO] is 0.75 ppb and the estimated net OH production rate from HONO photolysis is $1.0 \times 10^{-4}$ ppb s$^{-1}$. The comparison of OH production rates from $O_3$ and HONO photolysis reveals that HONO plays a more important role than $O_3$ in the wintertime AOC at the ground surface level of the Beijing urban area.

To investigate the contribution of HONO to the AOC and SOA formation, we have performed a sensitivity simulation in which the heterogeneous HONO sources are not considered and only homogeneous source of NO+OH is included (Hereafter referred to as HOMO case). Figure 4a shows the temporal profiles of the simulated HONO concentrations in the BASE and HOMO case compared with observations at IRSDE site from 9 to 26 January 2014. In the HOMO case with only the homogeneous reaction of NO and OH as the HONO source, the HONO concentrations are substantially underestimated against the observations, especially during nighttime, with a *MB* of -1.5 ppb. When the heterogeneous HONO sources of HONO are included in the BASE case, the model captures the temporal variation of HONO concentrations compared to the observations, with an *IOA* of 0.67, but it frequently underestimates the HONO concentration during nighttime. The HONO simulation results are generally consistent with previous studies, which demonstrate that the homogeneous source fails to interpret the observed high HONO concentrations and the heterogeneous HONO sources significantly improve the HONO simulations (e.g., Li et al., 2010).

Figure 4b shows the comparison of simulated SOA and observed OOA concentrations at IRSDE site. For the BASE case simulation, the model reasonably well reproduces the SOA temporal variation compared with observations, with an *IOA* of 0.81. It slightly underestimates the SOA concentration, with a *MB* of -0.4 μg m$^{-3}$, and the *RMSE* is rather large, around 9.8 μg m$^{-3}$, showing considerable deviations of the SOA simulation. When the heterogeneous HONO formation are excluded in the HOMO case, the model considerably underestimates the SOA concentration against the observations, with a *MB* of -3.2 μg m$^{-3}$. On average, the BASE case produces about 96% of the observed SOA concentrations, but only 65% for the HOMO case at IRSDE during the episode. Therefore, the SOA concentrations are substantially increased by the heterogeneous HONO sources, with an average SOA contribution of about 32% at IRSDE site. Obviously, the heterogeneous HONO

sources remarkably improve the SOA simulation, particularly during the heavy haze days. The SOA enhancement by the heterogeneous HONO sources in Beijing is not the same as the result in Mexico City (Li et al., 2010). Li et al. (2010) have showed that the heterogeneous HONO sources increase SOA concentrations by more than 100% in the morning in Mexico City but play a minor role during the rest of the day. In Beijing, the SOA enhancement due to the heterogeneous HONO sources is significant during the whole day. The main reason for the difference is that the high $O_3$ level in the afternoon

dominate the OH production in Mexico City. Additionally, the WRF-CHEM model also generally yields the observed HONO diurnal cycle, but the underestimation is substantial during nighttime (Figure 3b). The simulated $O_3$ diurnal cycle is in agreement with the observation at IRSDE site, but the model underestimates the $O_3$ concentration against the measurement in the morning (Figure 3b).

Figure 5 presents the comparison of simulated SOA and observed OOA diurnal cycles averaged during the episode at

260 IRSDE site. The observed SOA concentration continuously increases from the early morning (06:00 BJT) to the noon (12:00 BJT), due to the low PBL height and progressively increased photochemical production of SOA. After the noon, although the PBL commences to develop rapidly, the SOA concentration still increases until the evening (18:00 BJT), caused by the enhanced AOC to facilitate SOA formation. Compared to the HOMO case, the SOA diurnal cycle simulation is considerably improved in the BASE case against the measurement. The model with the heterogeneous HONO sources still fails to capture

the observed SOA peak during the evening and overestimates SOA concentrations against the measurement from 00:00 to 06:00 BJT, showing the WRF-CHEM model deficiency in simulating diurnal variation of SOA formation (Lennartson et al., 2018). It is worth noting that the heavy haze pollution in Beijing is generally markedly influenced by the regional transport (Wu et al., 2017; Li et al., 2018), so uncertainties in the wind field simulations have large potentials to affect the SOA diurnal cycle simulation (Bei et al., 2017).

The vertical distribution is an important feature for evaluating the climatic impact of OA. Previous studies have shown large discrepancies between the simulated SOA vertical distribution and aircraft measurements (Heald et al., 2011; Tsigaridis et al., 2014). Although the OA vertical distribution measurement is not available during the simulation episode, analyses are still performed to explore the difference in simulated vertical profiles of POA and SOA, caused by the heterogeneous HONO sources. Figure 6a shows the vertical distribution of the average simulated POA and SOA concentration during the episode

over IRSDE site in the BASE and HOMO case. POA and SOA concentrations decrease rapidly from the ground level to about 2 km, and are lower than 0.4 and 0.5 μg m$^{-3}$ above 2 km, respectively. The POA concentration at the ground level is much higher than that of SOA, but its decrease in vertical direction is by far faster than that of SOA, which is consistent with the observation in Beijing by Sun et al. (2015). They have found that the SOA contribution to the OA mass at 260 m is higher than that at the ground level. The SOA enhancement due to the heterogeneous HONO sources is remarkable near the

ground surface and rapidly decreases with the altitude, showing the dominant HONO contribution of the ground surface.

Generally, the heterogeneous HONO sources do not substantially increase the O$_3$ concentration in the PBL, with an enhancement of less than 4% (Figure 6b).

Figure 7a shows the spatial pattern of simulated near-surface SOA concentrations averaged during the episode in the BASE case. The high near-surface SOA concentrations are concentrated in the plain region of BTH, generally exceeding 10 μg m$^{-3}$, and can be up to 20 μg m$^{-3}$ in southern Hebei province. Figure 7b presents spatial distribution of the average near-surface SOA enhancement due to heterogeneous HONO sources ((BASE – HOMO)/HOMO*100). Heterogeneous HONO sources play an important role in the near-surface SOA formation, increasing the SOA concentrations by 10% to 55% in BTH. The SOA enhancement is remarkable in the plain region of BTH, more than 40%. The regional average near-surface SOA concentration is increased from 5.4 μg m$^{-3}$ in the HOMO case to 7.9 μg m$^{-3}$ in the BASE case by heterogeneous HONO sources, enhanced by about 46.3%.

### 3.3 SOA formation from different pathways in winter

Four SOA formation pathways are considered in the WRF-CHEM model, including (1) oxidation and partitioning of POA treated as semivolatile (PSOA), (2) oxidation of anthropogenic VOCs (ASOA), (3) oxidation of biogenic VOCs (BSOA), and (4) heterogeneous reactions of glyoxal and methylglyoxal on aerosol surfaces (HSOA). We have further analyzed the SOA formation from the four pathways in BTH during the episode.

Figure 8 shows the spatial distribution of the average predicted concentration of the near-surface PSOA, ASOA, BSOA, and HSOA during the whole simulation period. The BSOA concentration in BTH is rather low, less than 0.5 μg m$^{-3}$, caused by the low emissions of biogenic VOCs due to the weak insolation in winter. The spatial distributions of ASOA, PSOA, and HSOA are similar, showing similar emission patterns of their precursors. ASOA, PSOA, and HSOA are primarily distributed in the plain region of BTH, with the concentration exceeding 2.0, 10.0, and 6.0 μg m$^{-3}$ in the southern Hebei province, respectively.

Figure 9 provides the percentage contribution of ASOA, BSOA, PSOA, and HSOA to the total SOA mass averaged during the simulation period in BTH. PSOA dominates the total SOA mass in BTH, with a contribution of 58.9%. Unexpectedly, HSOA constitutes the second most important SOA formation pathway, contributing 27.6% to the SOA mass. The contribution of ASOA and BSOA is 11.6% and 1.9%, respectively. The average near-surface SOA mass concentration increases from 1.7 μg m$^{-3}$ in non-haze conditions (defined as hourly PM$_{2.5}$ concentration less than 75 μg m$^{-3}$) to16.1 μg m$^{-3}$ in haze conditions (defined as hourly PM$_{2.5}$ concentration exceeding 75 μg m$^{-3}$) (Figures 9b and 9c). The contribution of HSOA to the SOA mass increases from 8.5% in non-haze conditions to 30.2% in haze conditions, highlighting the importance of heterogeneous reactions of dicarbonyls to the SOA formation during haze days.

Considering that the irreversible uptake of glyoxal and methylglyoxal is an important pathway of SOA formation under haze conditions in BTH, the HSOA formation is further investigated. Sun et al. (2016) have resolved aqueous SOA (aq-SOA)

factors from the AMS measurement, and reported that the aq-SOA is correlated well with several specific fragment ions, including $C_2H_2O_2^+$ (m/z 58), $C_2O_2^+$ (m/z 56) and $CH_2O_2^+$ (m/z 46), which are typical fragment ions of glyoxal and methylgloxyal (Chhabra et al., 2010). Additionally, aq-SOA is also highly correlated with several sulfur-containing ions, e.g. $CH_3SO^+$, $CH_2SO_2^+$ and $CH_3SO_2^+$, which are typical fragment ions of methanesulfonic acid (MSA). Sulfate is also mainly formed in the aqueous phase during wintertime haze days (G. Li et al., 2017). $CH_2O_2^+$ (m/z 46) is not used to compare with the simulation, as it has the same m/z value with $NO_2^+$ ion, causing some biases. In addition, the concentrations of $CH_2SO_2^+$ cannot be extracted from the AMS measurement, so is not used for comparisons. Figure 10 shows the scatter plot of the simulated HSOA concentration and the AMS measured sulfate and several specific fragment ions concentration during the episode. The simulated HSOA exhibits good correlations with those specific fragment ions with correlation coefficients exceeding 0.50, especially with regard to the $C_2H_2O_2^+$ and $C_2O_2^+$ ions with correlation coefficients of 0.59 and 0.58, respectively, showing reasonable simulation of the HSOA formation. The correlation of sulfate with the HSOA is not as good as those of the fragment ions, indicating that non-heterogeneous sources also play a considerable role in the sulfate formation. All the correlations are statistically significant with p-value smaller than 0.01. Furthermore, the average observed OM/OC and O/C ratio during the simulation period are 1.42 and 0.21, respectively.

The gas-phase glyoxal and methylglyoxal are from direct emissions and secondary formations in the atmosphere. The residential living sources include biofuel and coal combustion, and attain peak emissions in winter for heating purposes in Northern China. M. Li et al. (2017) have estimated that residential sector contributes about 27% of non-methane VOCs emissions in 2010 in China and biofuel combustion contributes a large part of oxygenated VOCs, alkynes, and alkenes to residential sector emissions. Laboratory and field studies have shown that wildfires and agricultural waste burning also emit glyoxal and methylglyoxal. Hays et al. (2002) have detected glyoxal and methylglyoxal emissions from six kinds of biomass in US and measured their emission rates for different kinds of biomass. Zarzana et al. (2017) have observed glyoxal and methylglyoxal emissions from agricultural biomass burning plumes by aircraft. Koss et al. (2018) have measured the emission factors of glyoxal and methyglyoxal by burning biofuels characteristic of western US. Fu et al. (2008) have estimated that 20% of glyoxal comes from biomass burning and 17% from biofuel use on a global scale, and 5% and 3% of methylglyoxal comes from biomass burning and biofuel use, respectively. During wintertime, residential living emissions are the most important primary source of glyoxal and methylglyoxal in BTH. Figure 11 shows the spatial distribution of emissions of glyoxal and methylglyoxal from residential living sources. The intense emissions of glyoxal and methylglyoxal occur mainly in the plain region of BTH, and the high emission rates exceed $0.10 \times 10^6$ mole month$^{-1}$ and $0.05 \times 10^6$ mole month$^{-1}$, respectively. Glyoxal and methylglyoxal can also be produced from the oxidation of anthropogenic and biogenic VOCs, such as isoprene and aromatics (Fu et al., 2008; Myriokefalitakis et al., 2008).

To investigate the contribution of primary and secondary gas-phase glyoxal and methylglyoxal to HSOA, the HSOA formed from primary emissions and the oxidation of VOCs are marked as primary and secondary HSOA in the model,

respectively, and tracked in simulations. Figure 12a and 12b present the spatial distribution of the average concentration of primary HSOA and its contribution to the total SOA mass. The primary HSOA distribution well corresponds to the emissions of glyoxal and methylglyoxal in BTH and the primary HSOA mass concentrations exceed 5 µg m$^{-3}$ in the southern Hebei Province. The contribution of primary HSOA to the total SOA mass ranges from 20% to 40% in the plain region of BTH, and exceeds 40% in the western Shandong province, caused by the high emissions of glyoxal and methylglyoxal and the simulated low concentrations of PSOA, ASOA, and BSOA. The secondary HSOA concentrations are fairly low, less than 0.5 µg m$^{-3}$ in BTH, and its contribution to the total SOA mass does not exceed 4%, much lower than that of primary HSOA. The regional average of primary and secondary HSOA over BTH are 2.0 and 0.17 µg m$^{-3}$, contributing about 25.5% and 2.1% to the total SOA mass, respectively, showing that the primary HSOA constitutes an important SOA formation pathway.

It is worth noting that isoprene epoxydiol (IEPOX SOA) formed by aqueous chemistry also plays a considerable role in the SOA formation. However, Hu et al. (2017) have shown that, during the wintertime, the IEPOX SOA contribution to the SOA formation in BTH is insignificant due to the very low biogenic isoprene emissions and the elevated NO$_x$ concentrations which substantially suppress the production of IEPOX SOA from the isoprene oxidation.

## 4      Summary and Conclusions

In the present study, a heavy haze episode from 9 to 26 January 2014 in BTH is simulated using the WRF-CHEM model to investigate the impact of heterogeneous HONO sources on SOA formation and the SOA formation from different pathways. A previous study has shown that the model has generally well produced spatial distributions and temporal variations of PM$_{2.5}$, SO$_2$, NO$_2$, and O$_3$ concentrations when compared with observations during the episode in BTH (Li et al., 2018). The model also reasonably well captures the temporal variation of POA, HOA, BBOA+COA, and CCOA concentrations against the measurement in Beijing.

During the episode, the observed low O$_3$ concentration does not facilitate the OH production from the O$_3$ photolysis, and HONO becomes a dominant OH contributor in the surface level in Beijing. Model results reveal that when heterogeneous HONO sources are considered, the WRF-CHEM model reasonably reproduces the temporal variation of HONO concentrations against the measurement in Beijing. Heterogeneous HONO sources substantially enhance the SOA formation and also improve the SOA simulation. The regional average near-surface SOA concentration is increased by about 46.3% due to heterogeneous HONO sources during the episode.

The regional average contribution of ASOA, BSOA, PSOA, and HSOA to the total SOA mass are 11.6%, 1.9%, 58.9%, and 27.6% during the simulation period in BTH, respectively. HSOA constitutes the second most important contributor to the total SOA mass and the contribution increases from 8.5% in non-haze conditions to 30.2% in haze conditions, showing the importance of heterogeneous reactions of dicarbonyls to the SOA formation during haze days. In addition, glyoxal and

methylglyoxal emitted from residential living sources dominate the HSOA concentration, contributing about 25.5% to the total SOA mass on average, indicating that direct emissions of dicarbonyl compounds play an important role in the SOA formation during the wintertime haze days.

Our model results show that both the heterogeneous HONO sources and primary emissions of glyoxal and methylglyoxal play an important role in the SOA formation in BTH during the haze episodes, constituting the key factor to close the gap between measurements and simulations. It is worth to note that, although the simulated SOA is generally consistent with the measurement when heterogeneous HONO sources and irreversible uptake of dicarbonyl compounds are considered, SOA simulations are influenced by many factors, including measurements, meteorology, emissions, SOA formation mechanisms and treatments, which need to be investigated comprehensively.

*Acknowledgements*. This work is financially supported by the National Key R&D Plan (2017YFC0210000) and National Research Program for Key Issues in Air Pollution Control. Li Xing acknowledges the support by the National Natural Science Foundation of China (No. 41807310, 41661144020) and Shaanxi Province Postdoctoral Science Foundation (No. 2017BSHEDZZ61). The PSI authors acknowledge the financial support by the Swiss National Science Foundation (SNSF) within the project HAZECHINA (Haze pollution in China: Sources and atmospheric evolution of particulate matter, IZLCZ2_169986).

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

**Figure Captions**

Figure 1 WRF-CHEM model simulation domain with topography. Blue dot denotes the location of Institute of Remote
Sensing and Digital Earth (IRSDE site) in Beijing. Red dots denote centers of 22 cities over BTH with ambient
monitoring sites and the sizes of circles denote the number of ambient monitoring sites of cities.

Figure 2 Comparisons of observed (black dots) and simulated (solid red lines) diurnal profiles of near-surface hourly mass
concentrations of (a) POA, (b) HOA, (c) BBOA+COA, and (d) CCOA at IRSDE site in Beijing from 9 to 26 January
2014.

Figure 3 Diurnal cycle of observed (black line) and modeled (red line: BASE case; blue line: HOMO case) (a) $O_3$ and (b)
HONO concentrations averaged from 9 to 26 January 2014 at IRSDE site in Beijing.

Figure 4 Comparisons of observed (black dots) and simulated (solid red and blue lines for the BASE and HOMO cases,
respectively) diurnal profiles of near-surface hourly mass concentrations of (a) HONO and (b) SOA at IRSDE site in
Beijing from 9 to 26 January 2014.

Figure 5 Observed (black dots) and modeled (red line: BASE case; blue line: HOMO case) SOA diurnal cycle averaged from
9 to 26 January 2014 at IRSDE site in Beijing.

Figure 6 Vertical distribution of (a) SOA and POA and (b) $O_3$ concentrations averaged from 9 to 26 January 2014 at IRSDE
site in Beijing. Red line: BASE case; Blue line: HOMO case.

Figure 7 Spatial distribution of (a) the average SOA mass concentration for the BASE case and (b) the percentage SOA
enhancement due to the heterogeneous HONO sources during the simulation period.

Figure 8 Spatial distribution of the average SOA concentration from different formation pathways of (a) ASOA, (b) BSOA,
(c) PSOA, and (d) HSOA during the simulation period.

Figure 9 SOA contribution of different formation pathways over BTH (a) during the whole simulation period, (b) under
non-haze conditions, and (c) under haze conditions.

Figure 10 Scatter plot of the simulated HSOA concentration and the AMS measured $SO_4^{2+}$, $C_2H_2O_2^+$, $C_2O_2^+$, $CH_3SO^+$, and
$CH_3SO_2^+$ concentrations from 9 to 26 January 2014 at IRSDE site in Beijing. All the correlations are statistically
significant with $p$-value smaller than 0.01.

Figure 11 Spatial distribution of the emission rate of (a) glyoxal and (b) methylglyoxal from residential living sources in
January, 2014.

Figure 12 Spatial distribution of (a) average primary HSOA concentrations and (b) its contribution to the total SOA, and (c)
average secondary HSOA concentrations and (d) its contribution to the total SOA during the simulation period.


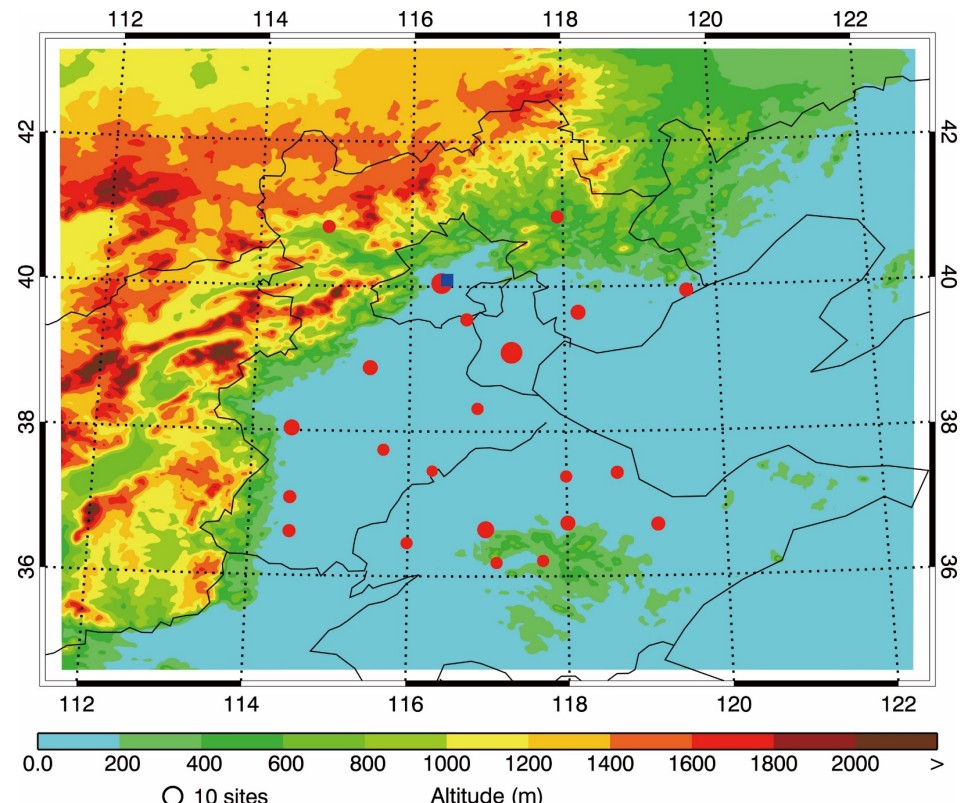

**Figure 1 WRF-CHEM model simulation domain with topography. Blue dot denotes the location of Institute of Remote Sensing and Digital Earth (IRSDE site) in Beijing. Red dots denote centers of 22 cities over BTH with ambient monitoring sites and the sizes of circles denote the number of ambient monitoring sites of cities.**



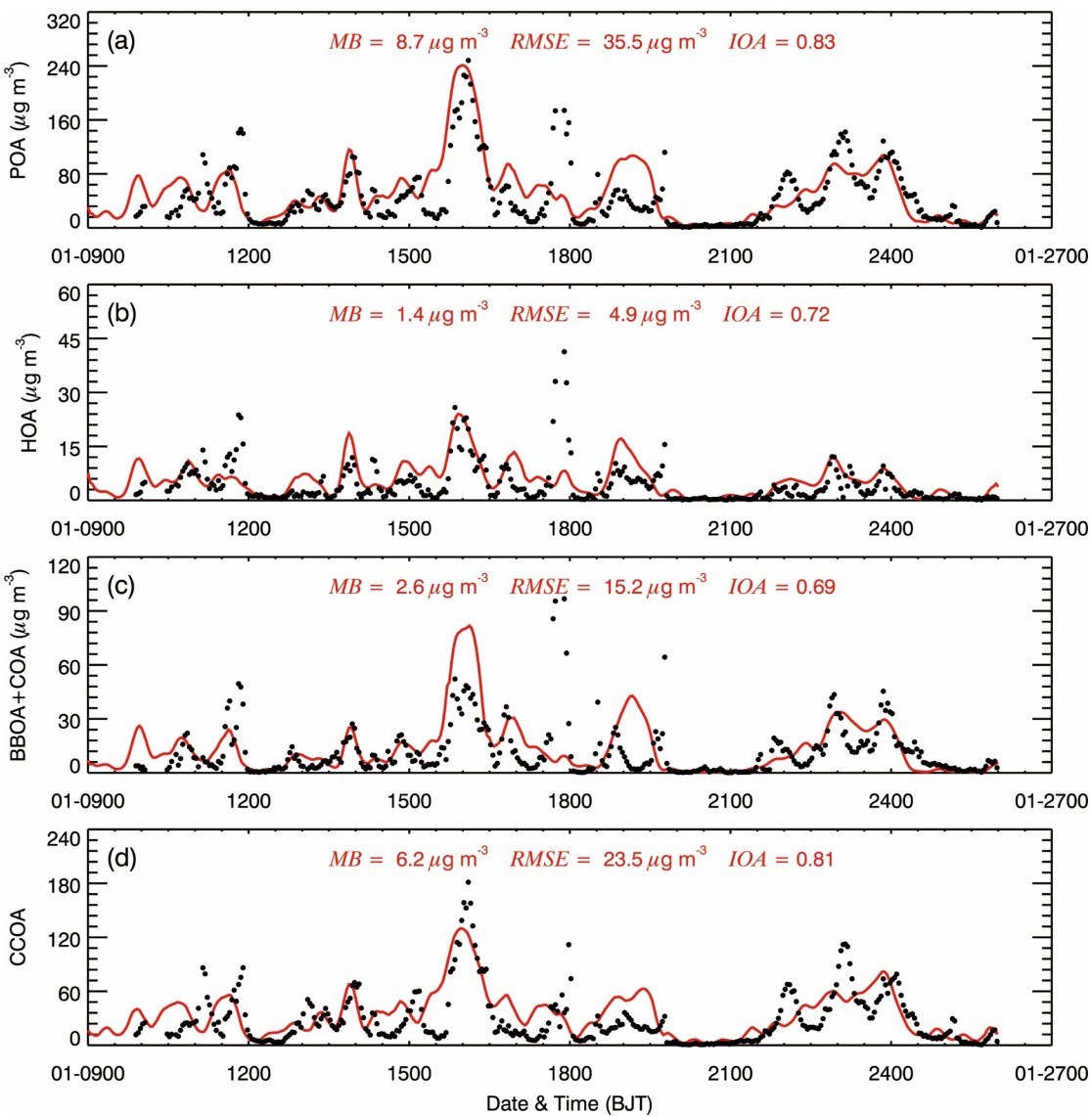

**Figure 2 Comparisons of observed (black dots) and simulated (solid red lines) diurnal profiles of near-surface hourly mass concentrations of (a) POA, (b) HOA, (c) BBOA+COA, and (d) CCOA at IRSDE site in Beijing from 9 to 26 January 2014.**

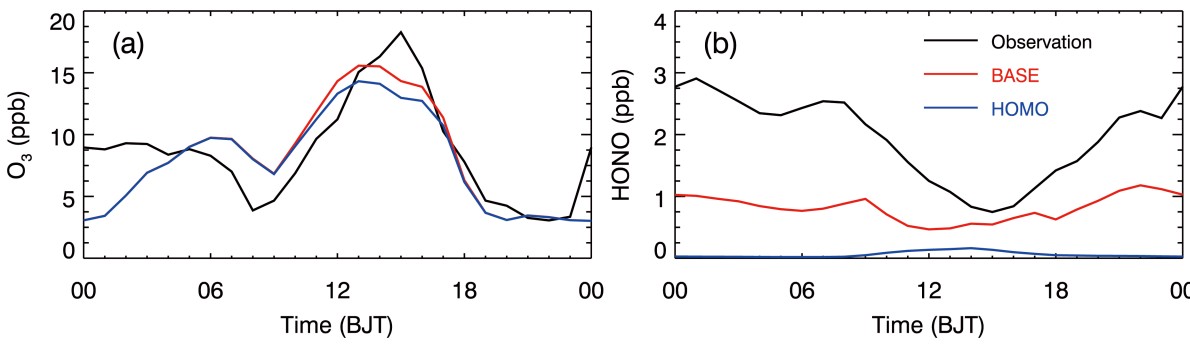

**Figure 3** Diurnal cycle of observed (black line) and modeled (red line: BASE case; blue line: HOMO case) (a) $O_3$ and (b) HONO concentrations averaged from 9 to 26 January 2014 at IRSDE site in Beijing.


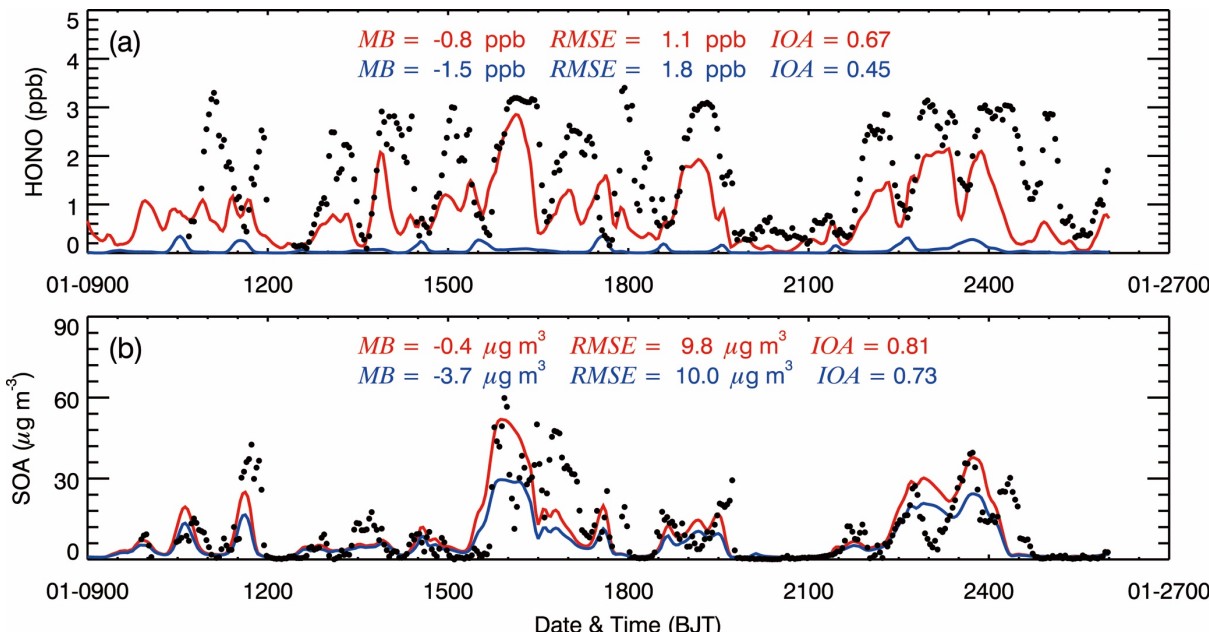

**Figure 4 Comparisons of observed (black dots) and simulated (solid red and blue lines for the BASE and HOMO cases, respectively) diurnal profiles of near-surface hourly mass concentrations of (a) HONO and (b) SOA at IRSDE site in Beijing from 9 to 26 January 2014.**

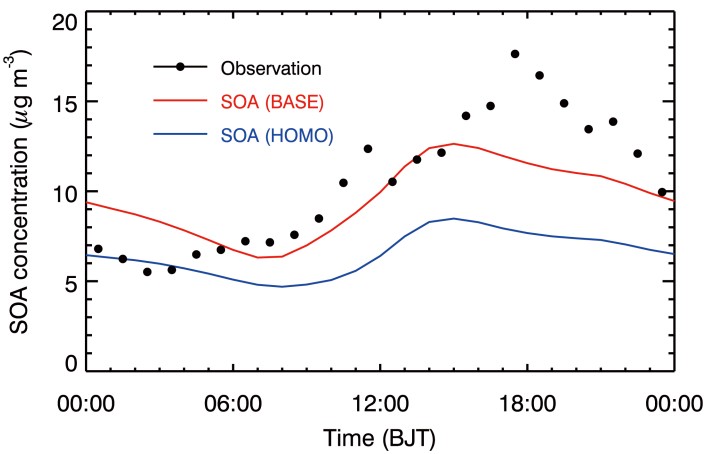

**Figure 5 Observed (black dots) and modeled (red line: BASE case; blue line: HOMO case) SOA diurnal cycle averaged from 9 to 26 January 2014 at IRSDE site in Beijing.**



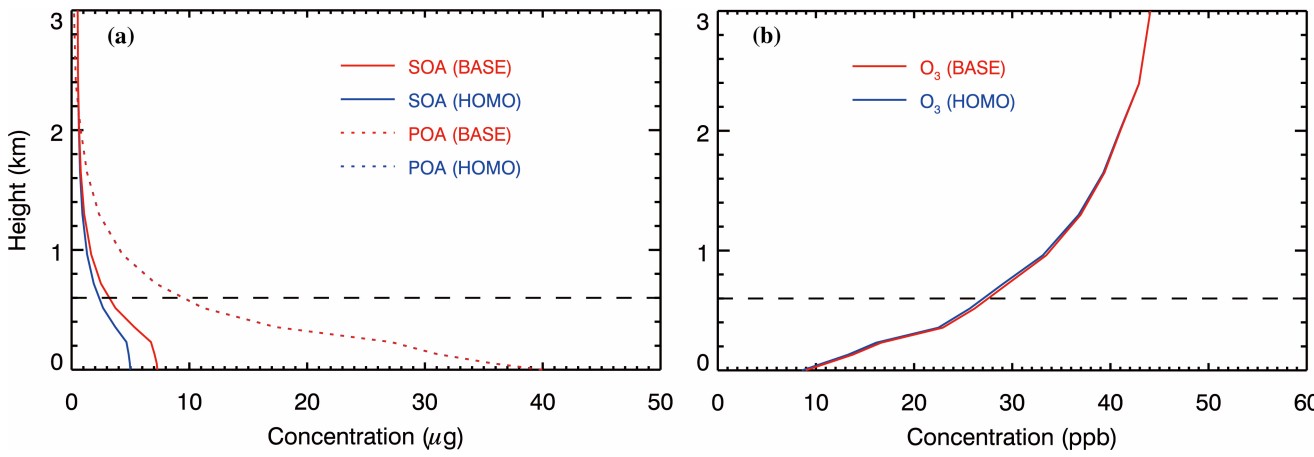


**Figure 6** Vertical distribution of (a) SOA and POA and (b) O$_3$ concentrations averaged from 9 to 26 January 2014 at IRSDE site in Beijing. Red line: BASE case; Blue line: HOMO case.


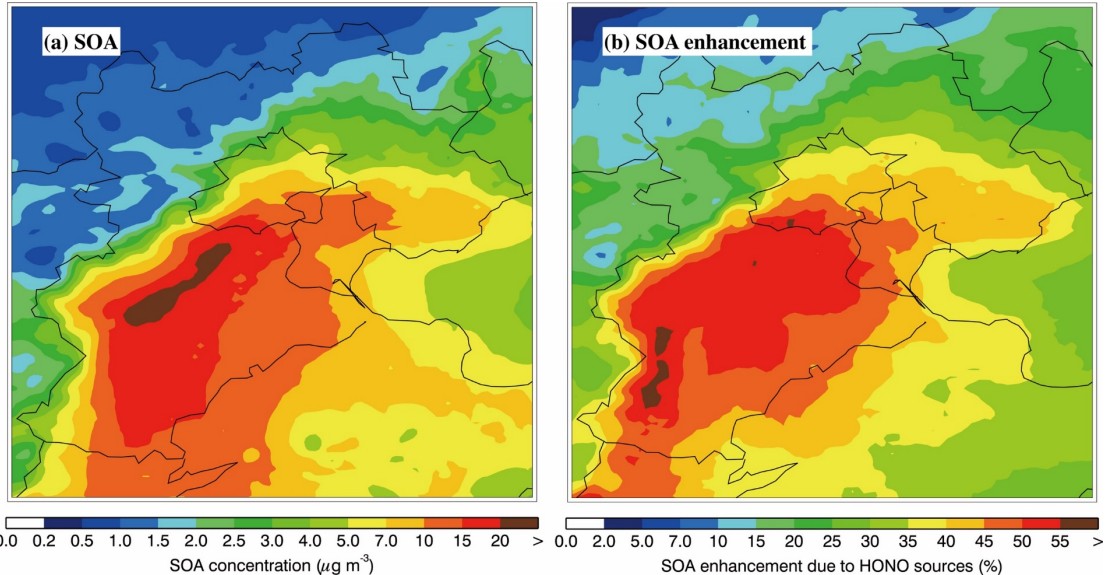

Figure 7 Spatial distribution of (a) the average SOA mass concentration for the BASE case and (b) the percentage SOA enhancement due to the heterogeneous HONO sources during the simulation period.


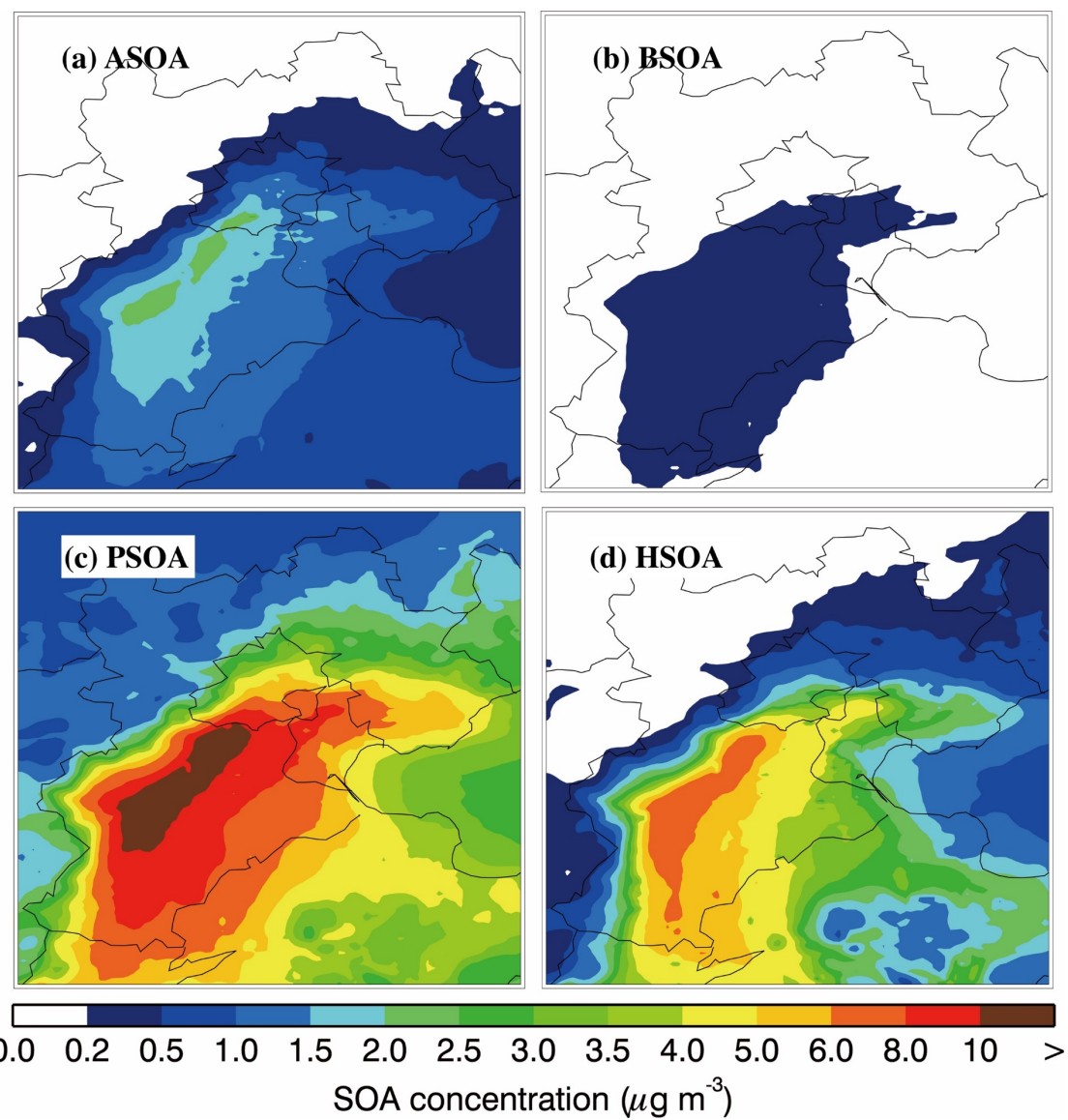

**Figure 8 Spatial distribution of the average SOA concentration from different formation pathways of (a) ASOA, (b) BSOA, (c) PSOA, and (d) HSOA during the simulation period.**


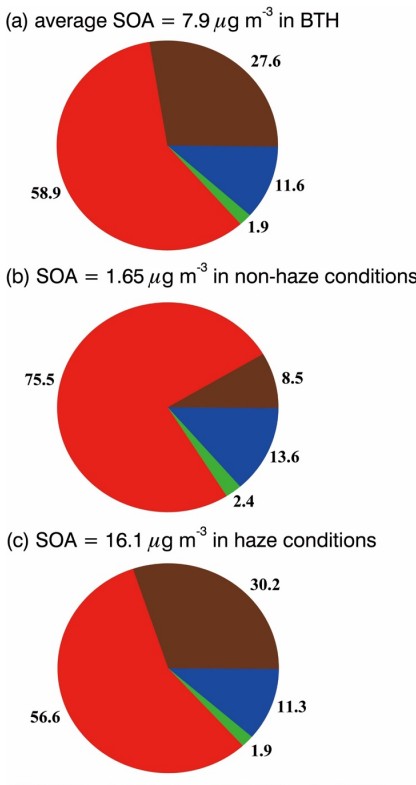

**Figure 9 SOA contribution of different formation pathways over BTH (a) during the whole simulation period, (b) under non-haze conditions, and (c) under haze conditions.**


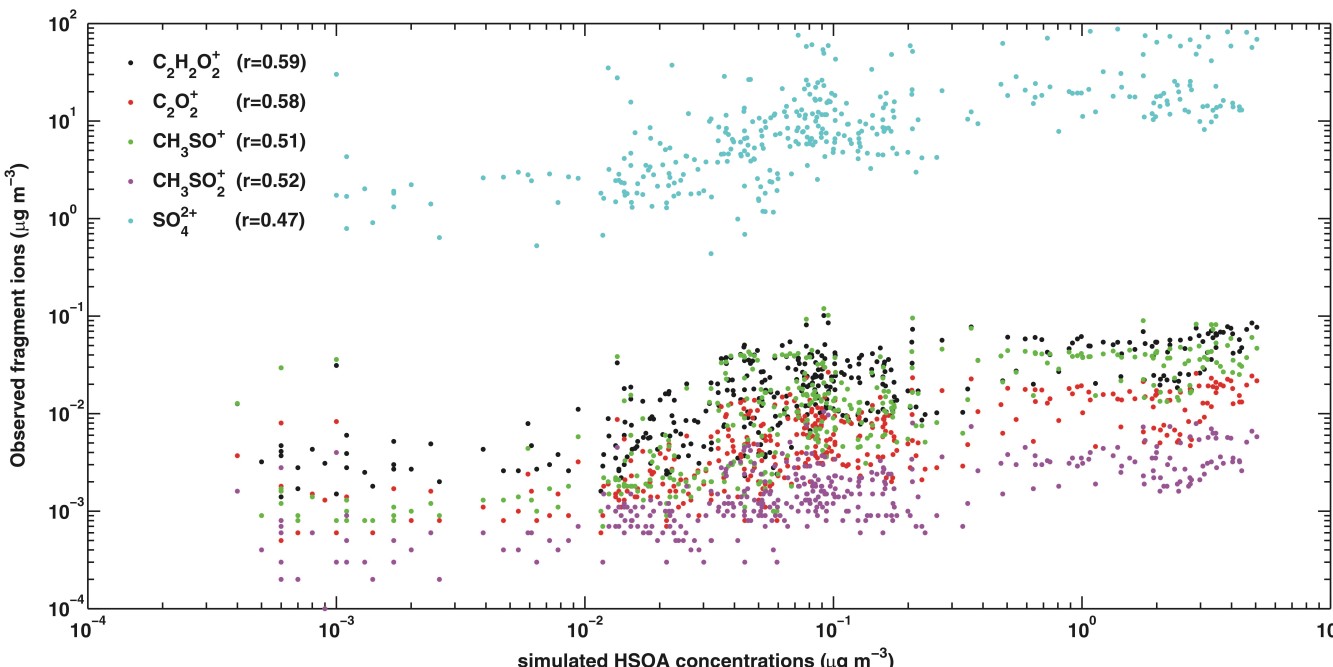


**Figure 10 Scatter plot of the simulated HSOA concentration and the AMS measured $SO_4^{2+}$, $C_2H_2O_2^+$, $C_2O_2^+$, $CH_3SO^+$, and $CH_3SO_2^+$ concentrations from 9 to 26 January 2014 at IRSDE site in Beijing. All the correlations are statistically significant with *p*-value smaller than 0.01.**


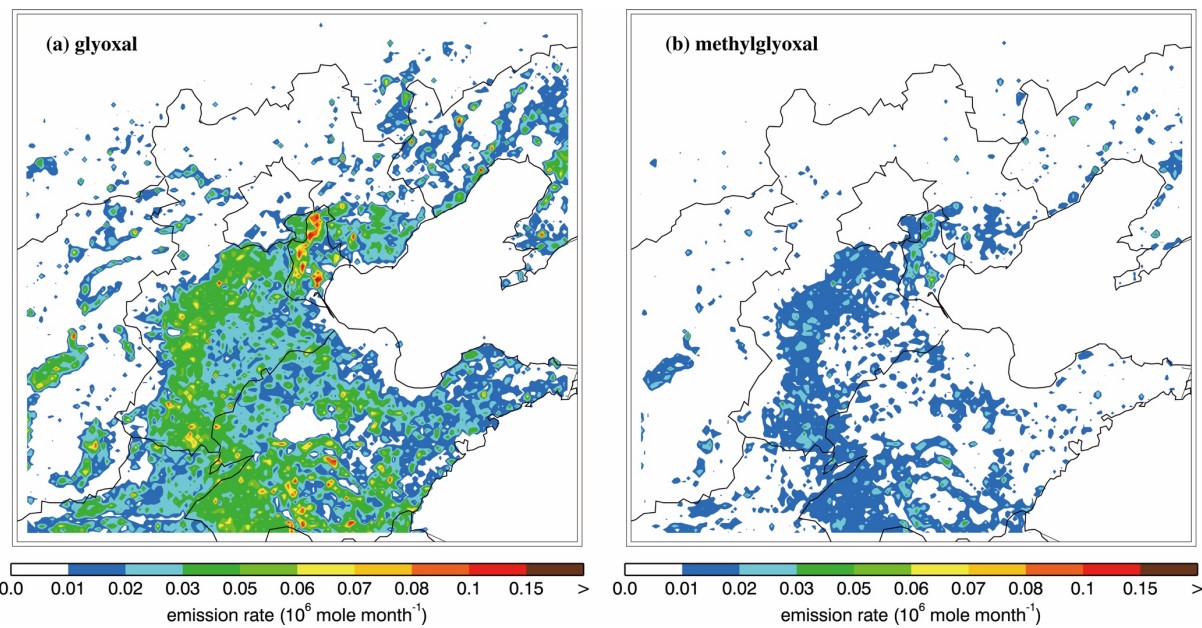

**Figure 11 Spatial distribution of the emission rate of (a) glyoxal and (b) methylglyoxal from residential living sources in January, 2014.**

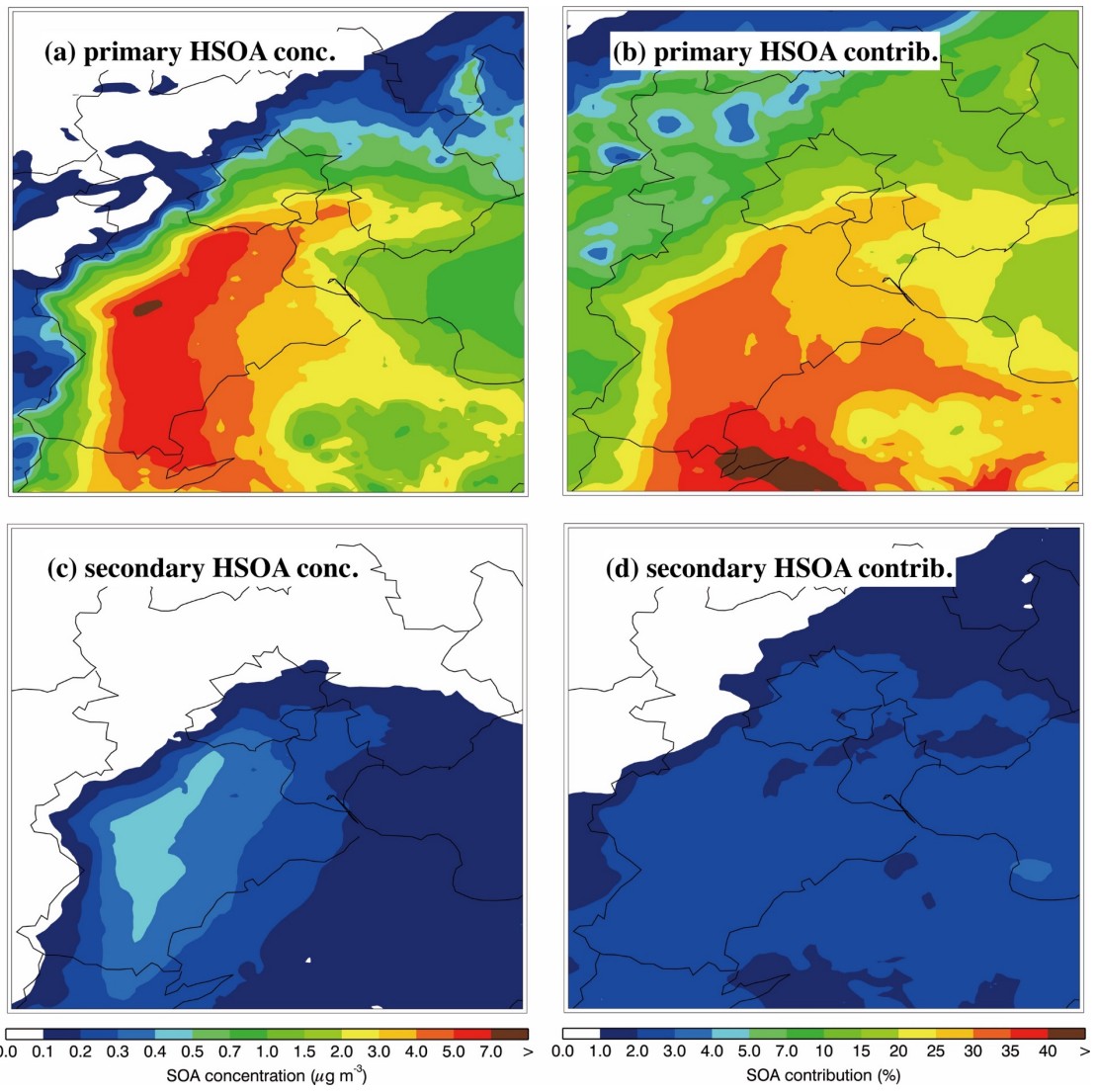

0.0  0.1  0.2  0.3  0.4  0.5  0.7  1.0  1.5  2.0  3.0  4.0  5.0  7.0  >
SOA concentration ($\mu$g m$^{-3}$)

0.0  1.0  2.0  3.0  4.0  5.0  7.0  10  15  20  25  30  35  40  >
SOA contribution (%)


**Figure 12 Spatial distribution of (a) average primary HSOA concentrations and (b) its contribution to the total SOA, and (c) average secondary HSOA concentrations and (d) its contribution to the total SOA during the simulation period.**
