# Peer review of "Wintertime secondary organic aerosol formation in Beijing-Tianjin-Hebei (BTH)"

_Atmospheric Chemistry and Physics, 2018_

## Referee Comment (RC1) · Anonymous Referee #2 · 3 Oct 2018

LiXing et al. present a modeling study of SOA formation pathways and contribution of heterogeneous HONO sources in the BTH region focusing on wintertime haze. This is an important study. I have several suggestions for strengthening the Manuscript, and I recommend that the following points need to be addressed before publication:

1. Introduction line 43: Please clarify if biogenic POA refers to POA from biomass burning and/or biological particles like bacteria, fungi etc.?

2. Line 46: In addition to Robinson and Hallquist, suggest citing the recent review paper on SOA by Shrivastava et al. 2017 (1)

3. Line 85: Also cite the global modeling paper using VBS by Shrivastava et al. 2015

(2)

4. Line 100: The difference between the 2-product model and VBS is also because the VBS accounts for semi-volatile and intermediate volatility organics emitted from fossil fuel and biomass burning sources in addition to traditional SOA 3. This needs to be mentioned here.

5. Line 115: Please mention the different SOA sources being represented by VBS: e.g., anthropogenic, biomass burning, biogenic. Also, was gas-phase fragmentation of organic vapors included during multigenerational aging of organic vapors in the VBS? Please see the following papers for reference: (References 2,4,5 listed below)

6. Section 3.1. POA simulations and Figure 2: It is confusing why POA and HOA are on separate panels. POA is generally compared against the HOA factor derived from PMF analysis of HR-Tof-AMS data. Please clarify what is model POA being compared to in panel 2(a) and what is PMF AMS HOA being compared to in panel 2(b)? Also I do not see a comparison for PMF OOA and model SOA.

7. Also, can the authors compare their glyoxal and methylglyoxal to some of the AMS factors? If not, can AMS total organic signal – (sum of HOA+BBOA+CCOA+COA) be used as an estimate of glyoxal/methylgloxal? If not, please explain. For example, are there any distinct AMS makers for glyoxal/methylgloxal SOA or aqueous SOA formed during Haze? What was the overall O:C ratio of AMS organic aerosol?

8. Section 3.3: The authors include glyoxal SOA, but seems they do not include isoprene epoxydiol (IEPOX SOA) which is also formed by aqueous chemistry. Is the IEPOX-SOA contribution expected to be insignificant?

9. Line 270-275: What sources contribute to residential living? Are these biofuel burning? Also, could glyoxal and methylglyoxal also be emitted from wildfires and agricultural waste burning?

10. Figure 3: In addition to observed average diurnal cycle, also include modeled

diurnal cycle average for O3 and HONO. For HONO please include both the base and HOMO cases from Figure 4.

References:

1 Shrivastava, M. et al. Recent advances in understanding secondary organic aerosol: Implications for global climate forcing. Rev. Geophys. 55, 509-559, doi:10.1002/2016RG000540 (2017).

2 Shrivastava, M. et al. Global transformation and fate of SOA: Implications of low-volatility SOA and gas-phase fragmentation reactions. J. Geophys. Res.-Atmos. 120, 4169-4195, doi:10.1002/2014jd022563 (2015).

3 Shrivastava, M., Lane, T. E., Donahue, N. M., Pandis, S. N. & Robinson, A. L. Effects of gas particle partitioning and aging of primary emissions on urban and regional organic aerosol concentrations. J. Geophys. Res.-Atmos. 113, doi:D18301 10.1029/2007jd009735 (2008).

4 Shrivastava, M. et al. Implications of low volatility SOA and gas-phase fragmentation reactions on SOA loadings and their spatial and temporal evolution in the atmosphere. J. Geophys. Res.-Atmos. 118, 3328-3342, doi:10.1002/jgrd.50160 (2013).

5 Shrivastava, M. et al. Sensitivity analysis of simulated SOA loadings using a variance-based statistical approach. Journal of Advances in Modeling Earth Systems 8, 499-519, doi:10.1002/2015ms000554 (2016).

---

## Referee Comment (RC2) · Anonymous Referee #1 · 22 Nov 2018

This paper presents WRF-Chem simulation results for a winter haze event in 9-26 January 2014 in Beijing-Tianjing-Hebei area. The results highlights the important effects of HONO, glyoxal, and methylglyoxal on SOA formation. The simulation results after considering all these effects yield significant improvement in comparison with observed SOA, O3, HONO, HOA, BBOA, and CCOA. The results are significant and the presentation is in high quality. This reviewer only has a few minor concerns before recommending it for publication.

1. While the introduction provides a good literature review of SOA importance and recent advances in modeling SOA, it is a bit open ended because it didn't address why

this paper is needed and how this paper differs from past studies.

2. It might be interesting to look a the difference in simulated vertical profiles of SOA and O3. Early studies with field campaigns in ACE-Asia and TRACE-P showed the model's deficiency in simulating vertical profiles of SOAs. For this study, such analysis could be purely a model analysis because there was no observational counterpart to compare. But, still this can be an interesting and adds more 'meat' in the paper.

3. Finally, is there any systematic diurnal variation of SOA? if so, how well model can simulate it? Diurnal variation of PM2.5 has been of a high interest for air quality studies and such variation may differ by region. Recent studies show that in east Asia, WRF-Chem has deficiency to capture the observed diurnal variation (see citation below and references therein). It is worthy to add some analysis or discussion in the paper. To what extent the SOA simulation is improved as far as diurnal variation is concerned?

Lennartson, E. et al., 2018: Diurnal variation of aerosol optical depth and PM2.5 in south Korea: a synthesis from AERONET, satellite (GOCI), KORUS-AQ observation, and WRF-Chem model, Atmospheric Chemistry and Physics, 18, 15125–15144.

---

## Author Comment (AC1) · 3 Jan 2019

**Reply to Anonymous Referee #1**

We thank the reviewer for the careful reading of the manuscript and helpful comments. We have revised the manuscript following the suggestion, as described below.

This paper presents WRF-Chem simulation results for a winter haze event in 9-26 January 2014 in Beijing-Tianjin-Hebei area. The results highlight the important effects of HONO, glyoxal, and methylglyoxal on SOA formation. The simulation results after considering all these effects yield significant improvement in comparison with observed SOA, $O_3$, HONO, HOA, BBOA, and CCOA. The results are significant and the presentation is in high quality. This reviewer only has a few minor concerns before recommending it for publication.

**1 Comment:** While the introduction provides a good literature review of SOA importance and recent advances in modeling SOA, it is a bit open ended because it didn't address why this paper is needed and how this paper differs from past studies.

**Response:** Previous studies have shown that CTMs are subject to underestimating SOA concentrations against measurements using the traditional two-product SOA module in China, particularly during wintertime haze days when the $O_3$ level is rather low. Therefore, in the study, we attempt to improve the SOA simulation based on a VBS SOA module, with focuses on the contribution of heterogeneous HONO sources and the uptake of glyoxal and methylglyoxal to the SOA formation during wintertime haze days in BTH. We have clarified in Section 1: "*Recent studies have demonstrated that CTMs are subject to underestimating SOA concentrations against measurements using the traditional two-product SOA module, particularly during wintertime haze days with rather low $O_3$ level (e.g., Jiang et al., 2012; Fu et al., 2012; Hu et al., 2017). Hence, it is imperative to improve the SOA simulations for supporting the design and implementation of emission control strategies to mitigate haze pollution in China.*

*In the study, the VBS SOA approach with aging implemented in the WRF-CHEM model is used to attempt to improve the SOA simulation during wintertime haze days in BTH, with focuses on the contribution of the heterogeneous HONO sources and the uptake of glyoxal and methylglyoxal to the SOA formation.*"

**2 Comment:** It might be interesting to look at the difference in simulated vertical profiles of SOA and $O_3$. Early studies with field campaigns in ACE-Asia and TRACE-P showed the model's deficiency in simulating vertical profiles of SOAs. For this study, such analysis could be purely a model analysis because there was no observational counterpart to compare. But, still this can be an interesting and adds more 'meat' in the paper.

**Response:** We have clarified in Section 3.2: "*The vertical distribution is an important feature for evaluating the climatic impact of OA. Previous studies have shown large discrepancies between the simulated SOA vertical distribution and aircraft measurements (Heald et al., 2011; Tsigaridis et al., 2014). Although the OA vertical distribution measurement is not available during the simulation episode, analyses are still performed to explore the difference in simulated vertical profiles of POA and SOA, caused by the heterogeneous HONO sources. Figure 6a shows the vertical distribution of the average simulated POA and SOA concentration during the episode over IRSDE site in the BASE and HOMO case. POA and SOA concentrations decrease rapidly from the ground level to about 2 km, and are lower than 0.4 and 0.5 µg m$^{-3}$ above 2 km, respectively. The POA concentration at the ground level is much higher than that of SOA, but its decrease in vertical direction is by far faster than that of SOA, which is consistent with the observation in Beijing by Sun et al. (2015). They have found that the SOA contribution to the OA mass at 260 m is higher than that at the ground level. The SOA enhancement due to the heterogeneous HONO sources is remarkable near the ground surface and rapidly decreases with the altitude, showing the dominant HONO contribution of the ground surface. Generally, the heterogeneous HONO sources do not substantially increase the $O_3$ concentration in the PBL, with an enhancement of less than 4% (Figure 6b).*"

**3 Comment:** Finally, is there any systematic diurnal variation of SOA? if so, how well model can simulate it? Diurnal variation of $PM_{2.5}$ has been of a high interest for air quality studies and such variation may differ by region. Recent studies show that in east Asia, WRF-Chem has deficiency to capture the observed diurnal variation (see citation below and references therein). It is worthy to add some analysis or discussion in the paper. To what extent the SOA simulation is improved as far as diurnal variation is concerned?

Lennartson, E. et al., 2018: Diurnal variation of aerosol optical depth and $PM_{2.5}$ in south Korea: a synthesis from AERONET, satellite (GOCI), KORUS-AQ observation, and WRF-Chem model, Atmospheric Chemistry and Physics, 18, 15125–15144.

**Response:** We have clarified in Section 3.2: "*Figure 5 presents the comparison of simulated SOA and observed OOA diurnal cycles averaged during the episode at IRSDE site. The observed SOA concentration continuously increases from the early morning (06:00 BJT) to the noon (12:00 BJT), due to the low PBL height and progressively increased photochemical production of SOA. After the noon, although the PBL commences to develop rapidly, the SOA concentration still increases until the evening (18:00 BJT), caused by the enhanced AOC to facilitate SOA formation. Compared to the HOMO case, the SOA diurnal cycle simulation is considerably improved in the BASE case against the measurement. The model with the heterogeneous HONO sources still fails to capture the observed SOA peak during the evening and overestimates SOA concentrations against the measurement from 00:00 to 06:00 BJT, showing the WRF-CHEM model deficiency in simulating diurnal variation of SOA formation (Lennartson et al., 2018). It is worth noting that the heavy haze pollution in Beijing is generally markedly influenced by the regional transport (Wu et al., 2017; Li et al., 2018), so uncertainties in the wind field simulations have large potentials to affect the SOA diurnal cycle simulation (Bei et al., 2017).*"

**References:**

Bei, N., Wu, J., Elser, M., Feng, T., Cao, J., El-Haddad, I., Li, X., Huang, R., Li, Z., Long, X., Xing, L., Zhao, S., Tie, X., Prévôt, A. S. H., and Li, G.: Impacts of meteorological uncertainties on the haze formation in Beijing–Tianjin–Hebei (BTH) during wintertime: a case study, Atmos. Chem. Phys., 17, 14579-14591, https://doi.org/10.5194/acp-17-14579-2017, 2017.

Fu, T. M., Cao, J. J., Zhang, X. Y., Lee, S. C., Zhang, Q., Han, Y. M., Qu, W. J., Han, Z., Zhang, R., Wang, Y. X., Chen, D., and Henze, D. K.: Carbonaceous aerosols in China: top-down constraints on primary sources and estimation of secondary contribution, Atmos. Chem. Phys., 12, 2725-2746, doi: 10.5194/acp-12-2725-2012, 2012.

Heald, C. L., Coe, H., Jimenez, J. L., Weber, R. J., Bahreini, R., Middlebrook, A. M., Russell, L. M., Jolleys, M., Fu, T.-M., Allan, J. D., Bower, K. N., Capes, G., Crosier, J., Morgan, W. T., Robinson, N. H., Williams, P. I., Cubison, M. J., DeCarlo, P. F., and Dunlea, E. J.: Exploring the vertical profile of atmospheric organic aerosol: comparing 17 aircraft field campaigns with a global model, Atmos. Chem. Phys., 11, 12673–12696, doi:10.5194/acp-11-12673-2011, 2011.

Hu, J., Wang, P., Ying, Q., Zhang, H., Chen, J., Ge, X., Li, X., Jiang, J., Wang, S., Zhang, J., Zhao, Y., and Zhang, Y.: Modeling biogenic and anthropogenic secondary organic aerosol in China, Atmos. Chem. Phys., 17, 77-92, doi: 10.5194/acp-17-77-2017, 2017.

Jiang, F., Liu, Q., Huang, X., Wang, T., Zhuang, B., and Xie, M.: Regional modeling of secondary organic aerosol over China using WRF/Chem, J. Aerosol. Sci., 43, 57-73, doi: 10.1016/j.jaerosci.2011.09.003, 2012.

Lennartson, E. M., Wang, J., Gu, J., Castro Garcia, L., Ge, C., Gao, M., Choi, M., Saide, P. E., Carmichael, G. R., Kim, J., and Janz, S. J.: Diurnal variation of aerosol optical depth and $PM_{2.5}$ in South Korea: a synthesis from AERONET, satellite (GOCI), KORUS-AQ observation, and the WRF-Chem model, Atmos. Chem. Phys., 18, 15125-15144, https://doi.org/10.5194/acp-18-15125-2018, 2018.

Li, X., Wu, J., Elser, M., Feng, T., Cao, J., El-Haddad, I., Huang, R., Tie, X., Prévôt, A. S. H., and Li, G.: Contributions of residential coal combustion to the air quality in Beijing–Tianjin–Hebei (BTH), China: a case study, Atmos. Chem. Phys., 18, 10675-10691, https://doi.org/10.5194/acp-18-10675-2018, 2018.

Sun, Y., Du, W., Wang, Q., Zhang, Q., Chen, C., Chen, Y., Chen, Z., Fu, P., Wang, Z., Gao, Z., and Worsnop, D.R.: Real-time characterization of aerosol particle composition above the urban canopy in Beijing: insights into the interactions between the atmospheric boundary layer and aerosol chemistry, Environ. Sci. Technol., 49(19), 11340-11347, doi: 10.1021/acs.est.5b02373, 2015.

Tsigaridis, K., Daskalakis, N., Kanakidou, M., Adams, P.J., Artaxo, P., Bahadur, R., Balkanski, Y., Bauer, S.E., Bellouin, N., Benedetti, A., Bergman, T., Berntsen, T. K., Beukes, J. P., Bian, H., Carslaw, K. S., Chin, M., Curci, G., Diehl, T., Easter, R. C., Ghan, S. J., Gong, S. L., Hodzic, A., Hoyle, C. R., Iversen, T., Jathar, S., Jimenez, J. L., Kaiser, J. W., Kirkevag, A., Koch, D., Kokkola, H., Lee, Y. H., Lin, G., Liu, X., Luo, G., Ma, X., Mann, G. W., Mihalopoulos, N., Morcrette, J. J., Müller, J. F., Myhre, G., Myriokefalitakis, S.,

Ng, N. L., O'Donnell, D., Penner, J. E., Pozzoli, L., Pringle, K. J., Russell, L. M., Schulz, M., Sciare, J., Seland, ø., Shindell, D. T., Sillman, S., Skeie, R. B., Spracklen, D., Stavrakou, T., Steenrod, S. D., Takemura, T., Tiitta, P., Tilmes, S., Tost, H., van Noije, T., van Zyl, P. G., von Salzen, K., Yu, F., Wang, Z., Wang, Z., Zaveri, R. A., Zhang, H., Zhang, K., Zhang, Q., and Zhang, X: The AeroCom evaluation and intercomparison of organic aerosol in global models, Atmos. Chem. Phys., 14, 10845-10895, doi: 10.5194/acp-14-10845-2014, 2014.

Wu, J., Li, G., Cao, J., Bei, N., Wang, Y., Feng, T., Huang, R., Liu, S., Zhang, Q., and Tie, X.: Contributions of trans-boundary transport to summertime air quality in Beijing, China, Atmos. Chem. Phys., 17, 2035-2051, https://doi.org/10.5194/acp-17-2035-2017, 2017.

[Figure]

Figure 5 Observed (black dots) and modeled (red line: BASE case; blue line: HOMO case) SOA diurnal cycle averaged from 9 to 26 January 2014 at IRSDE site in Beijing.

[Figure]

Figure 6 Vertical distribution of (a) SOA and POA and (b) O₃ concentrations averaged from 9 to 26 January 2014 at IRSDE site in Beijing. Red line: BASE case; Blue line: HOMO case.

---

## Author Comment (AC2) · 3 Jan 2019

**Reply to Anonymous Referee #2**

We thank the reviewer for the careful reading of the manuscript and helpful comments. We have revised the manuscript following the suggestion, as described below.

Li Xing et al. present a modeling study of SOA formation pathways and contribution of heterogeneous HONO sources in the BTH region focusing on wintertime haze. This is an important study. I have several suggestions for strengthening the Manuscript, and I recommend that the following points need to be addressed before publication:

**1 Comment:** Introduction line 43: Please clarify if biogenic POA refers to POA from biomass burning and/or biological particles like bacteria, fungi etc.?

**Response:** In the present study, biogenic POA refers to the POA from biomass burning. Biological particles like bacteria, fungi, pollen, and viruses are usually termed as primary biological aerosol particles, or bioaerosols. We have clarified in Section 1: "*POA are directly emitted into the atmosphere as particles by various anthropogenic and biomass burning sources*"

**2 Comment:** Line 46: In addition to Robinson and Hallquist, suggest citing the recent review paper on SOA by Shrivastava et al. 2017 (1)

**Response:** We have included the reference in Section 1: "*Some species of POA evaporate into the atmosphere and are oxidized further, re-partition into aerosols, and form SOA (Robinson et al., 2007; Hallquist et al., 2009; Shrivastava et al., 2017).*"

**3 Comment:** Line 85: Also cite the global modeling paper using VBS by Shrivastava et al. 2015.

**Response:** We have included the reference in Section 1: "*CTMs using the VBS approach have remarkably improved the SOA simulations against observations (e.g., Li et al., 2011; Shrivastava et al., 2013, 2015; Feng et al., 2016).*"

**4 Comment:** Line 100: The difference between the 2-product model and VBS is also because the VBS accounts for semi-volatile and intermediate volatility organics emitted from fossil fuel and biomass burning sources in addition to traditional SOA. This needs to be mentioned here.

**Response:** We have rephrased the sentence as suggested in Section 1: "*Additionally, the predicted ratio of secondary OC to total OC in the VBS approach is about 33%, much higher than that (around 5%) in the two-product model and also close to observation-based estimation (32%), suggesting a more realistic representation of the SOA formation by the VBS approach through accounting for the semi-volatile and intermediate volatility organics emitted from fossil fuel and biomass burning sources.*"

**5 Comment:** Line 115: Please mention the different SOA sources being represented by VBS: e.g., anthropogenic, biomass burning, biogenic. Also, was gas-phase fragmentation of organic vapors included during multigenerational aging of organic vapors in the VBS? Please see the following papers for reference: (References 2,4,5 listed below)

**Response:** We have rephrased the model description about the SOA formation as suggested in Section 2.1: "*The POA components from traffic-related combustion and biomass burning are represented by nine surrogate species with saturation concentrations (C\*) ranging from $10^{-2}$ to $10^6$ μg m$^{-3}$ at room temperature (Shrivastava et al., 2008), and assumed to be semi-volatile and photochemically reactive (Robinson et al., 2007). The SOA formation from each anthropogenic or biogenic precursor is calculated using four semi-volatile VOCs with effective saturation concentrations of 1, 10, 100, and 1000 μg m$^{-3}$ at 298 K. Previous studies have demonstrated that the fragmentation reactions of semi-volatile VOCs also play an important role in the SOA formation (Shrivastava et al., 2013, 2015, 2016). However, the fragmentation reactions have not been incorporated in the version of the WRF-CHEM model, and further studies need to been performed to include the contribution of those reactions to improve the SOA simulation.*"

**6 Comment:** Section 3.1. POA simulations and Figure 2: It is confusing why POA and HOA are on separate panels. POA is generally compared against the HOA factor derived from PMF analysis of HR-Tof-AMS data. Please clarify what is model POA being compared to in panel 2(a) and what is PMF AMS HOA being compared to in panel 2(b)? Also I do not see a comparison for PMF OOA and model SOA.

**Response:** Elser et al. (2016) have resolved five components of OA using a novel PMF method, which are HOA, BBOA, CCOA, COA, and OOA, respectively. HOA, BBOA, CCOA, and COA represent the POA related to the traffic combustion, biomass burning, coal combustion, and cooking emissions, respectively, and OOA is a surrogate for SOA. HOA in the study only represents the POA related to the traffic combustion emission, which is different from that used by Shrivastava et al. (2011), which represents the total urban POA and is not separated into different POA components. We have clarified in Section 2.3: "*Five components of OA are classified by their mass spectra and time series, including traffic-combustion hydrocarbon-like OA (HOA), cooking OA (COA), biomass burning OA (BBOA), coal combustion OA (CCOA), and oxygenated OA (OOA).*"

In Figure 2(a), the modeled POA is compared to the sum of HOA, BBOA, CCOA, and COA of AMS data resolved by PMF method. We have clarified in Section 2.1: "*Figure 2 presents the temporal profiles of the model-simulated and observed POA (sum of HOA, BBOA, CCOA, and COA), HOA, BBOA+COA, and CCOA concentrations from 9 to 26 January 2014 at IRSDE site in Beijing.*".

The comparison between modeled SOA and PMF-derived OOA is shown in Figure 4(b).

**7 Comment:** Also, can the authors compare their glyoxal and methylglyoxal to some of the AMS factors? If not, can AMS total organic signal – (sum of HOA+BBOA+CCOA+COA) be used as an estimate of glyoxal/methylgloxal? If not, please explain. For example, are there any distinct AMS makers for glyoxal/methylgloxal SOA or aqueous SOA formed during Haze? What was the overall O:C ratio of AMS organic aerosol?

**Response:** HOA, BBOA, CCOA, and COA are the primary OA, which cannot be used as an estimation of the glyoxal/methylglyoxal SOA. Sun et al. (2016) have resolved aqueous SOA factors from the AMS measurement and reported that the aqueous SOA (aq-SOA) is correlated well with several specific fragment ions, including $C_2H_2O_2^+$(m/z 58), $C_2O_2^+$ (m/z 56) and $CH_2O_2^+$ (m/z 46), which are typical fragment ions of glyoxal and methylgloxyal (Chhabra et al., 2010). In addition, aq-SOA is also highly correlated with several sulfur-containing ions, e.g. $CH_3SO^+$, $CH_2SO_2^+$ and $CH_3SO_2^+$, which are typical fragment ions of methanesulfonic acid (MSA), a secondary product from the oxidation of dimethyl sulfide (DMS). Also sulfate is

mainly formed in the aqueous phase during wintertime haze days (G. Li et al., 2017), which is compared with the simulated HSOA in Beijing. We have extracted the concentrations of those specific fragment ions reported in Sun et al. (2016) and compared with the simulated HSOA concentrations in Beijing. We do not use the concentrations of $CH_2O_2^+$ (m/z 46) for comparisons, as $NO_2^+$ ion has the same m/z value with $CH_2O_2^+$, which cause some biases. The concentrations of $CH_2SO_2^+$ cannot be extracted from the AMS data, which is not used for comparisons.

We have clarified in Section 3.3: "*Sun et al. (2016) have resolved aqueous SOA (aq-SOA) factors from the AMS measurement, and reported that the aq-SOA is correlated well with several specific fragment ions, including $C_2H_2O_2^+$(m/z 58), $C_2O_2^+$ (m/z 56) and $CH_2O_2^+$ (m/z 46), which are typical fragment ions of glyoxal and methylgloxal (Chhabra et al., 2010). Additionally, aq-SOA is also highly correlated with several sulfur-containing ions, e.g. $CH_3SO^+$, $CH_2SO_2^+$ and $CH_3SO_2^+$, which are typical fragment ions of methanesulfonic acid (MSA). Sulfate is also mainly formed in the aqueous phase during wintertime haze days (G. Li et al., 2017). $CH_2O_2^+$ (m/z 46) is not used to compare with the simulation, as it has the same m/z value with $NO_2^+$ ion, causing some biases. In addition, the concentrations of $CH_2SO_2^+$ cannot be extracted from the AMS measurement, so is not used for comparisons. Figure 10 shows the scatter plot of the simulated HSOA concentration and the AMS measured sulfate and several specific fragment ions concentration during the episode. The simulated HSOA exhibits good correlations with those specific fragment ions with correlation coefficients exceeding 0.50, especially with regard to the $C_2H_2O_2^+$ and $C_2O_2^+$ ions with correlation coefficients of 0.59 and 0.58, respectively, showing reasonable simulation of the HSOA formation. The correlation of sulfate with the HSOA is not as good as those of the fragment ions, indicating that non-heterogeneous sources also play a considerable role in the sulfate formation. All the correlations are statistically significant with p-value smaller than 0.01. Furthermore, the average observed OM/OC and O/C ratio during the simulation period are 1.42 and 0.21, respectively.*"

**8 Comment:** Section 3.3: The authors include glyoxal SOA, but seems they do not include isoprene epoxydiol (IEPOX SOA) which is also formed by aqueous chemistry. Is the IEPOX-SOA contribution expected to be insignificant?

**Response:** We have clarified in Section 3.3: "*It is worth noting that isoprene epoxydiol (IEPOX SOA) formed by aqueous chemistry also plays a considerable role in the SOA formation. However, Hu et al. (2017) have shown that, during the wintertime, the IEPOX SOA contribution to the SOA formation in BTH is insignificant due to the very low biogenic isoprene emissions and the elevated $NO_x$ concentrations which substantially suppress the production of IEPOX SOA from the isoprene oxidation.*"

**9 Comment:** Line 270-275: What sources contribute to residential living? Are these biofuel burning? Also, could glyoxal and methylglyoxal also be emitted from wildfires and agricultural waste burning?

**Response:** We have clarified in Section 3.3: "*The residential living sources include biofuel and coal combustion, and attain peak emissions in winter for heating purposes in Northern China. M. Li et al. (2017) have estimated that residential sector contributes about 27% of non-methane VOCs emissions in 2010 in China and biofuel combustion contributes a large part of oxygenated VOCs, alkynes, and alkenes to residential sector emissions. Laboratory and field studies have shown that wildfires and agricultural waste burning also emit glyoxal and methylglyoxal. Hays et al. (2002) have detected glyoxal and methylglyoxal emissions from six kinds of biomass in US and measured their emission rates for different kinds of biomass. Zarzana et al. (2017) have observed glyoxal and methylglyoxal emissions from agricultural biomass burning plumes by aircraft. Koss et al. (2018) have measured the emission factors of glyoxal and methyglyoxal by burning biofuels characteristic of western US. Fu et al. (2008) have estimated that 20% of glyoxal comes from biomass burning and 17% from biofuel use on a global scale, and 5% and 3% of methylglyoxal comes from biomass burning and biofuel use, respectively.*"

**10 Comment:** Figure 3: In addition to observed average diurnal cycle, also include modeled diurnal cycle average for $O_3$ and HONO. For HONO please include both the base and HOMO cases from Figure 4.

**Response:** We have included the modeled $O_3$ and HONO diurnal cycle in Figure 3 as suggested and clarified in Section 3.2: "*
[revised manuscript text omitted]